

# Multifaceted interventions to decrease mortality in patients with severe sepsis/septic shock—a quality improvement project

Brittany Siontis[1], Jennifer Elmer[2], Richard Dannielson[2], Catherine Brown[2], John Park[2], Salim Surani[3] and Kannan Ramar[4]

[1] Department of Internal Medicine, Mayo Clinic, Rochester, MN, United States
[2] Department of Pulmonary & Critical Care, Mayo Clinic, Rochester, MN, United States
[3] Division of Pulmonary, Critical Care & Sleep Medicine, Texas A&M University, Corpus Christi, TX, United States
[4] Division of Pulmonary and Critical Care, Mayo Clinic, Rochester, MN, United States

Corresponding author
Kannan Ramar,
ramar.kannan@mayo.edu

## ABSTRACT

Despite knowledge that EGDT improves outcomes in septic patients, staff education on EGDT and compliance with the CPOE order set has been variable. Based on results of a resident survey to identify barriers to decrease severe sepsis/septic shock mortality in the medical intensive care unit (MICU), multifaceted interventions such as educational interventions to improve awareness to the importance of early goal-directed therapy (EGDT), and the use of the Computerized Physician Order Entry (CPOE) order set, were implemented in July 2013. CPOE order set was established to improve compliance with the EGDT resuscitation bundle elements. Orders were reviewed and compared for patients admitted to the MICU with severe sepsis/septic shock in July and August 2013 (controls) and 2014 (following the intervention). Similarly, educational slide sets were used as interventions for residents before the start of their ICU rotations in July and August 2013. While CPOE order set compliance did not significantly improve (78% vs. 76%, $p = 0.74$), overall EGDT adherence improved from 43% to 68% ($p = 0.0295$). Although there was a trend toward improved mortality, this did not reach statistical significance. This study shows that education interventions can be used to increase awareness of severe sepsis/septic shock and improve overall EGDT adherence.

## INTRODUCTION

Aggressive and timely management of severe sepsis/septic shock is essential particularly with the increasing incidence (over one million cases projected in 2020 *Angus et al., 2001*), costs ($16.7 billion annually *Angus et al., 2001*), and burden of managing the morbidity and mortality. *Rivers et al. (2001)* showed the benefit of early goal-directed therapy (EGDT), with a decrease in overall mortality (46.9% vs. 30.5%) and length of hospital stay

**Table 1  Quality parameters in the elderly resuscitation bundle.**

| | |
|---|---|
| 1 | Lactate: Measured before or within 1 h after blood culture. |
| 2 | Blood culture: Drawn before antibiotics. |
| 3 | Antibiotic: Administered within 1 h of severe sepsis onset. |
| 4 | Fluid: Fluid given until one of the following<br>  a. CVP $\geq$ 8 (on MV 12) mmHg.<br>  b. MAP $\geq$ 65 mmHg and lactate < 2.5 mmol/L and UO > 0.5 mL/kgh.<br>  c. 12 L of crystalloid equivalent. |
| 5 | Vasopressor: Administered for 1 of the following<br>  a. MAP < 65 mmHg despite fluid challenge.<br>  b. MAP < 50 mmHg for $\geq$ 15 min. |
| 6 | RBC: Transfused if Hct < 30% and $ScVO_2$ < 70% or mixed venous O2 sat < 65% despite fluid resuscitation (RBC before adequate fluid resuscitation is inappropriate). |
| 7 | Inotrope: Started if Hct $\geq$ 30% and $ScVO_2$ < 70% or mixed venous O2 sat < 65% despite fluid resuscitation (inotrope before adequate fluid resuscitation is inappropriate). |

**Notes.**

CVP, Central Venous Pressure; MAP, Mean arterial pressure; RBC, Red blood cell; Hct, Hematocrit.

(18.4 vs. 14.6 days). The recent ARISE and ProCESS trials again confirm the importance of early aggressive management of patients with severe sepsis and septic shock (*Investigators et al., 2014*; *Mouncey et al., 2015*; *Pro et al., 2014*). However the role of all components of EGDT elements have been questioned. Despite multiple educational interventions from international societies and recommendations by the Surviving Sepsis Campaign (*Dellinger et al., 2004*; *Dellinger et al., 2013*) to institute resuscitation bundle elements in the management of severe sepsis/septic shock, all-or-none compliance with these bundle elements remain poor and the early recognition of sepsis remains a challenge (*Djurkovic et al., 2010*).

Various quality improvement interventions showed significant improvement in the all-or-none compliance with the resuscitation bundle elements and even more importantly, an improvement in mortality (*Schramm et al., 2011*). Schramm et al. implemented weekly feedback to care teams regarding their compliance in addition to starting a sepsis response team. Similarly, *Coba et al. (2011)* showed that monitoring the implementation of the resuscitation bundle elements by a continuous quality initiative, resulted in improvements in compliance and mortality.

Resident physicians play a significant role in the management of patients with severe sepsis/septic shock in our medical intensive care unit (MICU). Though our overall compliance with the resuscitation bundle elements in our MICU ranges from <50% to 80%, it could be consistently better. Resident physicians do not routinely receive data on the importance and elements of aggressive early resuscitation in patients with severe sepsis/septic shock. Also, a severe sepsis-specific Computerized Physician Order Entry (CPOE) that encompasses all of the resuscitation bundle elements is available to assist the physicians to comply with these elements (Table 1). The purpose of this quality improvement (QI) project was to identify barriers among resident physicians to comply with the resuscitation bundle elements, identify and implement interventions to improve compliance, and thereby reduce hospital/ICU LOS and 30 days mortality.

## METHODS

### Settings and participants

This QI project was conducted in the 24-bed MICU at Saint Mary's hospital, Rochester. Given the QI nature of the project, a waiver from the Institutional Review Board was obtained. All Internal Medicine (IM) residents were contacted via email giving a brief description of the QI problem statement and an attached survey. The QI problem statement to the residents stated the following: to decrease mortality of severe sepsis/septic shock to 10% in ICU patients by increasing compliance with early aggressive management of these patients using the resuscitation bundle elements by the use of the CPOE order set between 2013 and 2014 by the Internal Medicine residents. Their participation in the survey was voluntary.

### Intervention and comparison

CPOE is the only way to place orders at our institution. The components of the CPOE order set for severe sepsis/septic shock are listed in Table 2. All IM resident physicians rotating through the MICU during the phase of the QI project were surveyed to identify barriers to the use of CPOE severe sepsis order set (Table 3). Residents rotating through the MICU change at the beginning of every month with the number of residents rotating in MICU each month being the same. After identifying the barriers to successful compliance from the pre-intervention survey with the CPOE order set and resuscitation bundle elements, the week prior to starting the rotation, all residents were provided and educated with an education slide set that detailed the importance of early aggressive resuscitation of patients with severe sepsis/septic shock, and in using CPOE order sets to achieve compliance with the resuscitation bundle elements. The rationale for choosing education interventions along with feedback was based on the pre-survey results (in result section) and discussion among the team members using the priority grid matrix to identify interventions with high impact with low effort. The education interventions, along with feedback tools, were identified to be low effort with medium impact with minimal cost to implement. Compliance with resuscitation bundle elements was the major interested outcome. This slide set provided step-by-step instructions on how to access the order set. Pocket cards with criteria for using the order set were provided as an educational intervention, along with information regarding the resuscitation bundle element components were provided to every resident rotating through the ICU during the intervention period. Elements include time to antibiotics, obtaining cultures before antibiotic administration, lactate measurement, appropriate and timely volume resuscitation, inotrope and transfusion as appropriate (Table 1). Pocket cards also included the definition of the systemic inflammatory response syndrome (SIRS), sepsis, severe sepsis and septic shock to help residents identify those in need of early aggressive management of severe sepsis/septic shock. These cards were enlarged and placed on the roaming computers used during rounds and MICU admissions by each resident. Finally, residents were given a bi-monthly feedback sessions, compared to the pre-intervention once-monthly feedback sessions regarding their compliance with meeting CPOE order set and resuscitation bundle elements. At these

**Table 2  Components of the severe sepsis/septic shock management CPOE order set.**

**ALERT**

– Administer appropriate parenteral antibiotic within 1 h of sepsis recognition. The choice of antibiotics will depend on likelihood of specific infection, the patient immune status and allergies.

– Consider the following consults (if sepsis source known):

- Infectious Disease.
- General Surgery.
- Interventional Radiology.

– Activate Sepsis Response Team (if applicable to area) or appropriate resuscitation personnel is not available

Components of the order set checked by the provider:

1. Organ Perfusion:

  a. Obtain arterial blood gas every ____ hour(s) for ____ hours.

  b. Obtain central venous saturations (ScvO2 or SvO2) every _ *(1-2 h) place as guide under line*___ hour(s) for __ 6 (pre-filled)__ hours.

  c. Obtain Point Of Care serum lactate STAT. (should be a pre-checked box electronically).

  d. Obtain serum lactate every _ *(1-2 h) place as guide under line*___ hour(s) for __ 6 (pre-filled)__ hours. (should be a pre-checked box electronically)

2. Lab: Serum fasting glucose (not pre-checked).

3. Blood type and screen.

4. Vascular Access:

  a. Insert central line (do not have pre-checked).

5. Antibiotics

  a. (Various choices of antibiotics are listed and appropriate check boxes are present to be clicked).

6. Volume resuscitation: (At least 30ml/kg liters of fluid of one of the following).

  a. Lactated Ringers 1000 mL IV PRN over 15 min up to a maximum of _____ mL until one of the following are achieved:

  b. 0.9% NaCL 1000 mL IV PRN over 15 min up to a maximum of _______ mL or for 24 h until one of the following is achieved:

  c. Albumin 5% 500 mL IV PRN over 15 min up to a maximum of _____ mL until one of the following is achieved:

   i. To keep central venous pressure (CVP) at 12–15 mmHg (mechanically ventilated) or 8–12 mmHg (not mechanically ventilated).

   ii. Central Venous Pressure (CVP) $\geq$ 8 (on Mechanical Ventilation $\geq$ 12) mmHg.

   iii. MAP $\geq$ 65 mmHg and lactate <2.5 mmol/L and UO > 0.5 ml/kg/hr.

   iv. Lack of fluid responsiveness based on dynamic or static variables assessment.

7. Vasopressor infusion: Note: Recommend use only with central line, but in extreme emergency, vasopressors may be given for a brief period of time via peripheral site with constant monitoring for extravasation. Vasopressor should be administered for MAP <65 mmHg despite fluid challenge (30 ml/kg) (OR) MAP <50 mmHg for $\geq$ 15 min.

  a. Norepinephrine infusion 0.05 mcg/kg/minute, titrate by 0.05 mcg/kg/minute every 5 min to keep MAP $\geq$ 65 60–80 mmHg.

  b. Vasopressin 0.03 units/minute, do not titrate.

  c. Phenylephrine infusion 0.5 mcg/kg/minute, titrate by 0.1 mcg/kg/minute every 5 min to keep MAP $\geq$ 65 60–80 mmHg.

  d. Epinephrine infusion 0.05 mcg/kg/minute, titrate by 0.05 mcg/kg/minute every 5 min to keep MAP $\geq$ 65 60–80 mmHg.

8. Target ScVO2 $\geq$ 70 (or SvO2 less than 65%) and downward trending Lactate towards normal values by considering (one or more of the following):

  a. If ScvO2 less than 70% or SvO2 less than 65% (decreased oxygen delivery in spite of adequate volume replacement and preload):

   i. Dobutamine infusion 5 mcg/kg/minute titrate by 2.5 mcg/kg/minute every 10 min up to a maximum of 15 mcg/kg/minute to keep ScvO2 greater than 70% or SvO2 greater than 65%.

   ii. Milrinone 0.375 mcg/kg/minute titrate up to a maximum of 0.75 mcg/kg/minute to keep ScvO2 greater than 70% or SvO2 greater than 65%.

  b. If anemia present, consider transfusing packed red blood cells for a hemoglobin level less than 710 mg/dL.

**Table 3  Pre- and post intervention survey questions.**

| Question | Answer choices |
|---|---|
| Indicate year of training | PGY-1<br>PGY-2<br>PGY-3 |
| Number of months spent in MICU | 0 months<br>1 month<br>2 months<br>>2 months |
| Are you familiar with the severe sepsis order set in MICS? | Yes<br>No |
| Were you knowledgeable/aware of when to and when not to use the order set? | Yes<br>No |
| Did you have occasions when you later realized you should have instituted the severe sepsis order set? | Yes<br>No |
| What factors prevented you from using the severe sepsis order set? (please select all that apply) | Forgot<br>Didn't think it applied<br>Burdensome to use order set<br>Did not know how to access order set<br>Did not think order set had all elements needed |
| What factors are likely to promote the increased use of the severe sepsis order set? (Please select all that apply) | Easier accessibility in MICS<br>Demonstration on how to access order set<br>Reminders from seniors/fellows/staff to use the order set |
| * Post intervention questions only<br>Which among the below interventions has helped you the most to comply with the severe sepsis order set? | Educational interventions<br>Bimonthly feedback to the team<br>Reminders posted on the computers<br>All of the above |
| * Post intervention questions only<br>How have the above interventions helped? | Improve CPOE order set compliance<br>Increased knowledge and awareness of severe sepsis/septic shock<br>Increased awareness to be compliant with the resuscitation bundle elements<br>All of the above |
| * Post intervention questions only<br>While in the MICU, have you been using the severe sepsis/septic shock CPOE order set? | Always<br>Most of the time<br>Some of the time<br>Rarely |

sessions, residents were again reminded on the importance of early aggressive management of severe sepsis/septic shock and compliance with the resuscitation bundle elements, along with the use of the CPOE order set for all patients admitted with severe sepsis/septic shock. The compliance was checked by the physician data entry in the computerized system. Additionally, residents were given compliance data and feedback on the resuscitation bundle elements for patients admitted during their service time who met criteria for severe sepsis/septic shock in order to identify situations in which the order set should have been used.

The intervention was evaluated with a pre-post- test study design. To assess baseline compliance, patients admitted to the MICU with severe sepsis/septic shock in July

and August 2012 were identified. All IM residents rotating through the MICU were administered the initial survey within one week of starting their ICU rotation. Similarly, the post survey administration happened within 1 week of the IM residents starting their ICU rotation following the interventions. The number of IM residents rotating through each month remained the same. Patients qualified as having severe sepsis/septic shock if systolic blood pressure remained <90 mmHg despite adequate fluid resuscitation, lactate >4 mmol/L, or organ dysfunction/failure ensued due to hypoperfusion attributable to sepsis. Overall compliance with CPOE order set and the resuscitation bundle elements were determined by reviewing orders placed for patients admitted to the MICU with severe sepsis/septic shock. The interventions were implemented on July 1, 2013. Compliance with CPOE order set use and resuscitation bundle elements for patients admitted in July and August 2013 were assessed for comparison. Following the intervention period, the survey was re-administered to the IM residents who had rotated through the MICU, with additional questions addressing which interventions were beneficial in improving compliance.

## Outcomes and data collection

The MICU sepsis group keeps a database of patients admitted with severe sepsis/septic shock. Patients with severe sepsis were initially identified by screening criteria using the sepsis alert software in our MICU. Our quality coach nurses subsequently screened these patients to confirm the diagnosis of severe sepsis/septic shock before the data were manually entered in the MICU database. The team leaders performed periodic checks to check the validity. This database was used to identify patients in our timeframe of interest and to assess compliance. Once identified, orders placed for each patient were reviewed. Use of the CPOE order set was recorded, as well as whether 100% of the resuscitation bundle elements were met. Our experienced and ICU trained nurses were the quality coaches who collected all the relevant outcome data. The two nurses who were the quality coaches underwent rigorous training in data collection and analysis. They had a trial run of collecting and analyzing data in the QI project that was subsequently supervised by a quality expert in ICU for reliability and validity before they started this project. The two nurses conducted and collected the pre and post intervention data. Though the quality coach nurses were not blinded, the consistency of collection of data were reliable and valid as periodic checks were performed independently by the MD quality expert. Demographic data, outcomes including MICU and hospital length of stay, Acute Physiology and Chronic Health Evaluation II (APACHE II) score, Sequential Organ Failure Assessment (SOFA) score and mortality were also collected by our quality coaches.

## Statistical analysis

Statistical differences in patient demographics, CPOE compliance, resuscitation bundle elements compliance and 30-day mortality were compared between pre- and post-intervention groups using a chi-square model. Statistical differences between median hospital and ICU length of stay (LOS) were compared using Mann–Whitney test using R Statistical Software (*R Development Core Team, 2010*).
## RESULTS

### Survey results

First and third year residents rotated in our MICU. In the pre-intervention period, 56 of 170 IM residents who had rotated in our MICU during the QI project, participated in the survey with 31 (55%) of respondents being first year residents. In our MICU, it is the first year residents who are primarily responsible for order entry. The majority of residents (89%) were familiar with the order sets, however only 67% felt knowledgeable about when to use the order set. Additionally, 63% of residents identified at least one situation in which they later realized the order set applied to their patient. Uncertainty as to whether the CPOE order set applied to a particular patient with severe sepsis/septic shock (45% of respondents) was the largest barrier to order set compliance according to survey results from IM residents (Fig. 1A). They identified reminders from staff as the most likely factor to promote order set use (Fig. 1B).

In the post-intervention period, 44 of 170 IM residents who rotated in our MICU participated in the survey with 41% of respondents being first year residents. The greatest barrier to order set use remained uncertainty about whether it applied to their patient (36% of respondents). Again, reminders from staff were considered to be the most likely factor to improve compliance. The post-intervention survey contained questions directed at the interventions themselves. Educational interventions (23%) and bimonthly feedback (23%) were found to be most useful, while 25% of residents felt all of the interventions were equally helpful in improving compliance. Thirty percent of residents felt the interventions increased knowledge and awareness of severe sepsis/septic shock, while 11% found that the interventions increased awareness about the resuscitation bundle elements and thus increased CPOE order set compliance. Twenty seven percent of residents found increased awareness of both severe sepsis/septic shock and the resuscitation bundle elements from these interventions. Finally, 66% of residents reported using the CPOE order set always or most of the time, while only 9% reported rarely using the order set.

There were 51 patients admitted to the MICU for severe sepsis/septic shock in the pre-intervention period and 41 patients in the post-intervention period. Baseline demographics are shown in Table 4. There were no significant differences in age or BMI. The pre-treatment group had a higher percentage of males (59%) while the post-intervention period had more female admissions (56%); however these were not statistically different (Table 4).

Pre-intervention CPOE compliance was 78% while post-intervention compliance was 76% ($p = 0.74$). Compliance with meeting 100% of the resuscitation bundle elements was 43% in the pre-intervention and improved significantly to 68% in the post-intervention period ($p = 0.0295$). The median hospital LOS pre-intervention was 7.43 (range 3.85–16.09) days and decreased to 5.54 (range 3.31–9.62) days post-intervention ($p = 0.11$, Table 4). The median MICU LOS was 2.03 (1.34–3.83) day pre-intervention and decreased to 1.55 (0.92–2.96) days post intervention ($p = 0.085$). The 30-day mortality was 25% in the pre-intervention period and 12% in the post-intervention period ($p = 0.14$) (Table 5).

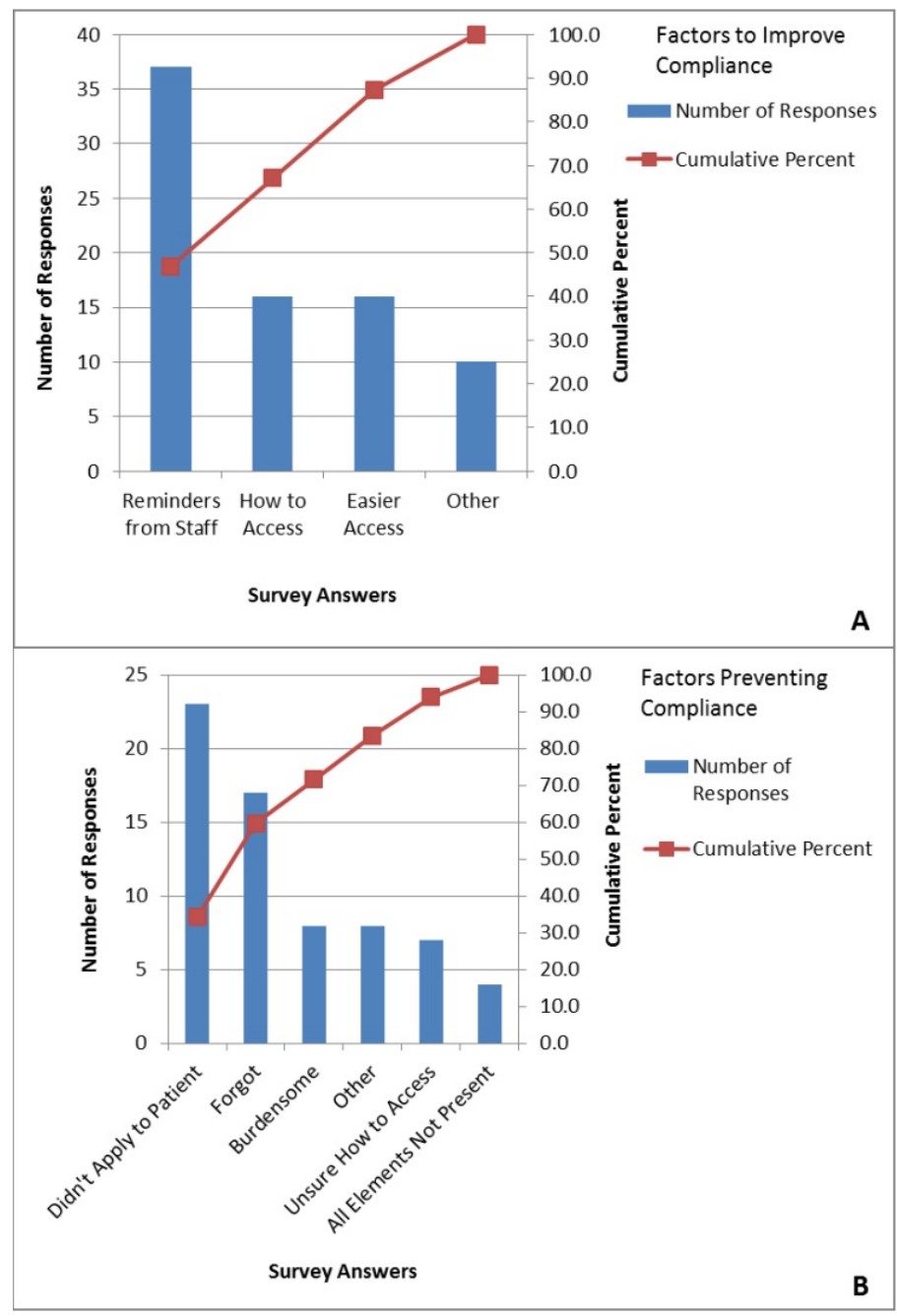

**Figure 1 Barriers based on survey of internal medicine residents.**

## DISCUSSION

Our QI initiative that used multifaceted educational and feedback interventions based on the identified barriers, successfully improved the overall compliance with the resuscitation bundle elements (43–68%, $p = 0.0295$), decreased the ICU and hospital length of stay. These findings were reached despite a lack of significant improvement in the CPOE order

**Table 4 Demographics.**

|  | Pre-intervention ($N = 51$) | Post-intervention ($N = 41$) | *P* value |
|---|---|---|---|
| Mean age Years (SD) | 66 (13.7) | 68 (16.3) | 0.61 |
| Gender N (%) | F 21 (41) M 30 (59) | F 23 (56) ) M 18 (44) | 0.09 |
| BMI Mean (SD) | 30.7 (9.26) | 29.2 (6.39) | 0.46 |
| APACHE Mean (SD) | 85.5 (26.9) | 78.2 (29.0) | 0.19 |
| SOFA Mean (SD) | 7.61 (4.17) | 7.49 (4.13) | 0.89 |

**Table 5 Outcomes.**

|  |  | Pre-intervention ($N = 51$) | Post-intervention ($N = 41$) | *P* value |
|---|---|---|---|---|
| Hospital LOS Median (IQR) |  | 7.43 (3.85–16.09) | 5.54 (3.31–9.62) | 0.11 |
| MICU LOS Median (IQR) |  | 2.03 (1.34–3.83) | 1.55 (0.92–2.96) | 0.85 |
| Mortality N(%) | 30 day | 13 (25) | 5 (12) | 0.14 |

set compliance, emphasizing the important of education, feedback and overall increasing the awareness of early aggressive management of patients with severe sepsis and septic shock among resident physicians.

The primary barrier identified though the resident survey was the lack of understanding on when to use the CPOE order set. We suspect that this lack of understanding stemmed from a deficiency of knowledge regarding the definition of severe sepsis/septic shock and what parameters are used to define and identify these patients. The education intervention not only provided information regarding early aggressive management with the use of resuscitation bundle elements and its importance, but also definitions from SIRS to severe sepsis/septic shock. Post-intervention surveys confirmed the increased awareness of when to use the order set.

Several studies have investigated educational interventions to improve compliance with the resuscitation bundle elements as outline by the Surviving Sepsis Campaign guidelines (*Levy et al., 2010*; *Nguyen et al., 2007*; *Schramm et al., 2011*). A prospective study of severe sepsis in 54 ICUs in Spain noted an improvement of overall compliance with the sepsis resuscitation bundle from 5.3% to 10% based on educational interventions (*Ferrer et al., 2008*). Our study also shows that educational interventions alone can improve compliance with meeting the resuscitation bundle elements. In addition to educational interventions, this quality improvement project also provided bi-monthly feedback to residents on their overall compliance. The study in Spain focused on all-or-none compliance, while our study focused on order set compliance. The educational interventions in our study were

similar to those in Spain, which provided pocket cards, posters and educational slides with definitions of severe sepsis and septic shock, appropriate management, and periodic feedback on performance (*Ferrer et al., 2008*). Their intervention also included providing educational materials to emergency department and surgical physicians.

*Nguyen et al. (2007)* showed increased compliance with the resuscitation bundle elements from 0% to 51.2% using educational interventions in addition to feedback on a quarterly basis. Our study differs in that feedback was provided on a bi-monthly basis. Their study also differed in that interventions were initiated in the Emergency Department, while our interventions were in the MICU. With quarterly feedback to nurses and physicians, they saw an increase in sepsis resuscitation bundle elements compliance from zero to 51.2% at the end of two years. They noted no change in ED LOS or hospital LOS between patients with and without bundle elements completed.

Though, all-or-none compliance improved significantly post intervention in our study, the compliance with the use of the CPOE order set did not improve. There are various reasons to explain this discrepancy. It is likely that some of the resuscitation bundle elements were being done prior to the patient's admission to the MICU, particularly in the Emergency department (ED). For example, many patients have central lines placed and initial laboratory evaluations done in the ED. In those situations, some find it easier to place individual orders for elements still needed to best coordinate timing of repeat labs rather than going through the entire order set.

Additionally, some elements of the CPOE order set may be omitted if the resident did not feel that particular element was necessary. This was particularly true for patients who did not have a central access for ScvO2 monitoring. The unchecking of certain elements of the CPOE order set would have then resulted in non-compliance with the resuscitation bundle elements. Finally, practitioners often have their own method of approaching patient management. Some prefer to think about each element of the resuscitation bundle elements individually rather than ordering them as a whole. This would be unlikely to change with our interventions, and therefore still contributes to reduced compliance.

A study by *Rubenfeld (2004)* categorized reasons for the discordance between guidelines and practice into three groups: knowledge barriers, attitude barriers and behavioral barriers (*Rubenfeld, 2004*). While our intervention was knowledge focused, perhaps the most difficult to address, and likely the cause for ongoing imperfect compliance, is attitude barriers. We can hope that through continued education, these attitudes will change.

*Rivers et al. (2001)* found mortality of the control group to be 45.6% compared to 30.5% in the EGDT group. Additionally, a study by *Lin et al. (2006)* showed a mortality rate of 71.6% in the control group compared with 53.7% in the EGDT group. Compared with these trials, our baseline mortality of 25% is lower, and more in line with the outcomes of recent trials, which showed 18–21% mortality (*Investigators et al., 2014*; *Pro et al., 2014*). For several years, our institution has stressed the importance of identifying patients with severe sepsis/septic shock and meeting EGDT standards. Several QI projects, such as that by *Schramm et al. (2011)*, have been aimed at this mission, contributing to our low baseline mortality.

There are several limitations to this study. This is a single-centered study with a distinct organization and staffing, making generalizability difficult. Additionally, the pre- and post-intervention time periods were only two months. The sample sizes of the patients that were studied were small, which contributed to the lack of statistical significance in some of the outcome measures.

## CONCLUSION

In conclusion, we have shown that a multifaceted intervention strategy of educational intervention to our resident physicians to increase awareness of early aggressive management using the resuscitation bundle elements in patients with severe sepsis and septic shock, along with continued feedback on performance on these measures, resulted in significant improvements in all or none compliance with resuscitation bundle elements and trended toward improved mortality among severe sepsis and septic shock patients in our MICU. This method and success can be applied towards improving the attitude and behavioral changes towards other disease specific order sets. We plan to sustain this improvement with continued feedback along with the educational intervention.

### Funding
The authors received no funding for this work.

### Competing Interests
The authors declare there are no competing interests.

### Author Contributions
- Brittany Siontis conceived and designed the experiments, performed the experiments, analyzed the data, contributed reagents/materials/analysis tools, wrote the paper, prepared figures and/or tables, reviewed drafts of the paper.
- Jennifer Elmer and Richard Dannielson conceived and designed the experiments, performed the experiments, analyzed the data.
- Catherine Brown conceived and designed the experiments, performed the experiments, analyzed the data, wrote the paper, reviewed drafts of the paper.
- John Park conceived and designed the experiments, performed the experiments, analyzed the data, contributed reagents/materials/analysis tools, wrote the paper.
- Salim Surani wrote the paper, prepared figures and/or tables, reviewed drafts of the paper.
- Kannan Ramar conceived and designed the experiments, performed the experiments, analyzed the data, wrote the paper, prepared figures and/or tables, reviewed drafts of the paper.

## Ethics

The following information was supplied relating to ethical approvals (i.e., approving body and any reference numbers):

Approved by the IRB of Mayo Clinic, Rochester, MN USA. Since this is a QI project, an IRB exemption was obtained.

## Supplemental Information

Supplemental information for this article can be found online at http://dx.doi.org/10.7717/peerj.1290#supplemental-information.

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
