# Peer review of "Multifaceted interventions to decrease mortality in patients with severe sepsis/septic shock—a quality improvement project"

_PeerJ, doi:10.7717/peerj.1290_

## Round 0.1 · original submission · Major Revisions

As reviewer 2 has indicated a number of important points that affect major conclusions of the study, in your revised submission please provide point-to-point rebuttal of issues that have been identified by the reviewer.

·

Basic reporting

Concurs with standards

Experimental design

The CPOE order set (use of which is one of the 2 primary outcomes of the study) is insufficiently explained. The authors should contribute a line detailing what is contained in this order set.
Statistical analysis – it is unclear why the authors chose to use an ANOVA for the comparison of the medians of two independent groups (ICU stay and hospital LOS pre and post intervention). A Mann Whitney U-test would possibly be a more valid test to compare medians in this case.

Validity of the findings

The authors have shown an increased compliance with EGDT bundles in their institution but not with CPOE ordering. However the EDGT compliance is a more significant outcome in any case as the CPOE ordering may or may not represent a “tick-box” exercise (this is unclear from the manuscript). They have stated that “Our study shows a decrease in mortality with increased EGDT compliance, suggesting there are still opportunities to decrease mortality even with a low mortality at baseline”. This is a completely misleading statement and needs to be removed. There was no decrease in mortality, ICU LOS or hospital LOS demonstrated in this study. This was a small QI study and is certainly not adequately powered to demonstrate a mortality benefit from the intervention assessed.

Additional comments

This was a small likely well-executed study on the importance of education for early intervention in sepsis for junior doctors working a large tertiary ICU. It did demonstrate that with simple tools such as pocket cards that there was an increased awareness among this group and an improved compliance with EGDT. The significance of this should not be overstated however and there was no improvement in mortality and one would not be expected in any case in a study of this size in a unit that already has a low mortality rate. I would also speculate that this intervention led to increased awareness of the importance of EGDT amongst the wider group of staff working in this ICU (cards were left on the portable computers). This may have also accounted for the improvement in compliance shown in this case.

Reviewer 2 ·

Basic reporting

Dykstra et al. present a single-center quality improvement project evaluating multifaceted interventions (provision of an educational slide set and pocket cards to resident physicians as well as increasing feedback sessions from monthly to bi-monthly) to decrease mortality in patients with severe sepsis/septic shock in their MICU.
The paper begins with an introduction to the problem of severe sepsis/ septic shock and a discussion of research in the area of quality improvements aiming to increase compliance with resuscitative guidelines. The problem in the Authors’ own unit is described, whereby there is an inconsistent wide range of compliance with resuscitation bundle elements. The current situation is described whereby resident physicians (who play a significant role in the management of patients with sepsis/septic shock) do not routinely receive information on the importance of and the elements of aggressive early resuscitation. There is a computerized order entry system that is designed to assist the physicians to comply with each of the elements of their resuscitation bundle.
The purpose of the project is stated in the text and summarized by me as follows:
Purpose 1. To identify barriers among resident physicians to comply with EGDT resuscitation bundle elements.
Purpose 2. To identify interventions to improve compliance
Purpose 3. To implement interventions to improve compliance
Purpose 4. To reduce hospital/ICU LOS and mortality by implementing the above interventions
Overall the introduction describes the problem and the rationale for developing the quality improvement project as well as the purpose(s) of the project.
I have the following suggestions:
I would remove the word “mismanagement” from line 85. The costs associated with sepsis are very high due to the condition itself. Our aim is to to recognize sepsis earlier and to institute treatment earlier in an attempt to decrease mortality and morbidity. There is controversy surrounding treatment and so the word “mismanagement” may not be entirely appropriate where management does not follow EGDT as described by Rivers (the SSC bundles have since been updated to reflect the results the more recent studies).
The sentence at line 87 is reasonable and does not refer to EGDT but rather early management (a reference to the recently published ProMISe trial should probably be added here – findings essentially in line with the the ARISE and ProCESS trials).
The term EGDT has become synonymous with Rivers and formed the basis for much of the resuscitation management in the Surviving Sepsis Campaign guidelines. However, EGDT is not exactly the same as the SCC bundles. I would probably separate/define the use of the terms “SCC bundles” and EGDT (and “resuscitation bundle elements” and “aggressive early resuscitation” referred to later in the text) and decide on the best term to use for the resuscitation bundle being used in the Authors’ unit. There should be consistency in the use of terms and a clear definition of the resuscitation bundle used in this institution should be presented in the paper.
Line 88/89 – references required for “multiple educational interventions from international societies”
Line 108 Mortality timeframe is not described (30-day mortality is described later at line 158)

Experimental design

Methods

Settings and Participants
The QI project was conducted in one MICU. An IRB approval waiver was received. A QI problem statement was emailed to all IM residents. A survey was also sent by email prior to residents rotating through the MICU – resident participation was voluntary.
I would suggest that the wording of the QI problem statement be included in the paper.
I have some questions regarding the survey:
Residents rotate on a 1-monthly basis yet all IM residents were surveyed. How many of the whole group rotated through MICU during the 2-month period (presumably no more that 1/6 assuming 1 month MICU per year per year of residency)? Were these the 1/6 that responded to the survey? It is reasonable that the opinions of all residents would be sought when trying to determine barriers to compliance prior to the intervention period however it would seem that sending the survey to all IM residents to determine answers to post-intervention questions would yield very little useful data unless the residents had actually rotated through the MICU during the July & August study timeframe. Even so, how would it be possible to residents to answer the post-intervention –specific questions if they had not yet been exposed to the interventions? This is all very unclear and requires further explanation
Intervention and Comparison
Were there residents that didn’t place orders in the CPOE? Is this group different in some way to the group described as “all IM residents” This is important to know and should be consistent.
It is unclear when the survey (Table 1) was administered. Was it before starting the rotation or during the rotation? If it was before starting the rotation for the first time then it is unlikely to have been possible to answer many of the questions (e.g. “Did you have occasions……” This needs to be made clearer.
In the survey, residents could be anywhere from PGY-1 to PGY-3. It would be important to know if the distribution of residents was similar between the pre-intervention and post-intervention groups (especially since the data appears to have been collected). When comparing compliance between groups it is important that the groups are similar, otherwise, any differences in outcomes may be due to differences in the groups rather than due to the intervention.
Purpose 1. Identify barriers among resident physicians to comply with EGDT resuscitation bundle elements.
A survey was emailed to residents (Table 1). This dealt with the issues pertaining to completing the CPOE order set only. This is not the same as identifying barriers to compliance with “resuscitation bundle elements” (i.e. what was actually done, not what was ordered to be done).
Perhaps goal number 1 needs to be revised to reflect this?

Purpose 2. Identify interventions to improve compliance
Although some of the barriers to complying with the CPOE order set were identified above it is unclear how interventions to improve compliance were identified. For example, it is not stated why or how the provision of the educational slide set was identified as an intervention to improve compliance. Similarly, the same applies to the provision of pocket cards and the increase in frequency of feedback sessions. Perhaps this goal should be removed or at least an explanation should be offered as to why the interventions that were chosen were chosen.


The following interventions were put in place:

A Provision of an educational slide set to residents (prior to the implementation of this intervention residents did not receive this slide set)
1 detailing the importance of early aggressive resuscitation of patients with severe sepsis/septic shock.
2 detailing the importance of using CPOE order sets to achieve compliance with EGDT measures
3 providing step-by-step instructions on how to access the order set

B Provision of pocket cards (prior to the implementation of this intervention residents did not receive pocket cards)
1 detailing criteria for using the order set (I presume this means when and for which patients the order set was to be used)
2 detailing information regarding EGDT components
3 defining SIRS, sepsis, severe sepsis and septic shock to help “identify those in need of EGDT”
These cards were also enlarged and placed on computer stations used during
rounds and during MICU admissions

C Residents were given bi-monthly feedback sessions (prior to the implementation of this intervention residents were given monthly feedback sessions)

The interventions (A, B & C above) was evaluated with a pre-post- test study design.

Purpose 3. Implement interventions to improve compliance
The first part of this “Implement interventions” may or may not have been achieved. We do not know if the slide sets or pocket cards were received by all residents or whether they actually received by monthly feedback sessions or not. It is important that this information be presented.


Patients with severe sepsis/septic shock were identified for the study periods (Pre- July & August 2012; post – July & August 2013)
Line 140. It is unclear how patients with severe sepsis/septic shock were identified. I am not sure how this would be possible using computerized data only. I can only assume that the database referred to on line 150 is how the patients were actually identified for the purpose of this study. Please clarify.

Line 142. There is a big difference between assessing for compliance with completing an order set and assessing compliance with physically performing each of the resuscitation elements. Just because an order was entered doesn’t mean that the order was carried out. Conversely, a clinical intervention could have been completed without being reflected in a particular order/order set record. The methodology used here needs to be presented

Outcomes and Data Collection
Line 153. I am very unclear. I understand how it might be pretty straight-forward to identify whether order sets were completed for a group of patients. I would need a better explanation of the process for determining whether there was compliance in meeting each of the EGDT bundle elements. How was this actually done in practice – by determining on a timeline when fluids were given and over what timeframe and by assessing records of hemodynamics? To my mind this allows a lot of flexibility in determination and therefore introduces bias through an un-blinded data analyzer.

Purpose 4. To reduce hospital/ICU LOS and mortality by implementing the above interventions
With such low numbers of patients it is unlikely that any difference in any of these outcome measures listed could possibly reach statistical significance.

Validity of the findings

Results
Survey Results
I am not sure what to make of the survey results. It is unclear why all residents were included (presumably even those that had not even rotated through MICU yet given that about half of the respondents were PGY-1s and the survey took place in July & August (i.e. the beginning of the year)). I would like to see all the data not just the data for questions 6 and 7 to determine the demographics of the group. In addition, if the survey was sent out prior to the intervention time-frame then most of the questions referred to a period before the intervention timeframe and those marked with an asterisk could not have been answered as the interventions had not yet been implemented. Maybe I am missing something here – either way the reader needs a clearer explanation. If the survey was administered prior to the rotation, how could the resident assess the interventions?
It would be interesting to know why each of the interventions was chosen based on each of the barriers identified.
There was no difference in CPOE compliance between timeframes
There was a significant difference in bundle compliance between timeframes. Again, the reader needs to know how this was assessed and whether the assessor was blinded (I cannot see how he/she could have been blinded based on the date being present on each order set/vital sign measurement in the electronic patient record). Who was the assessor? This is likely to be a fundamental flaw with the design of the project.
Where the difference in a particular variable between two groups does not reach significance then it cannot be said that there is a difference between groups for that variable. Therefore, there was no difference in median hospital LOS, ICU LOS or 30-day mortality in this study and it is important to state this clearly.
Table 2
Patient groups appear similar based on the (reasonable) demographic factors chosen.

Additional comments

I commend the Authors for performing this QI project. It makes empiric sense that interventions such as the ones described in this paper would lead to improved compliance with guidelines and therefore to better outcomes for patients. Proving this is quite another thing, however. Proving that an intervention makes a difference to an outcome requires meticulous attention to detail in designing any trial or QI project. It is hugely important that incorrect conclusions are not drawn due to results that have been influenced by the introduction of unintentional biases.
Although I am confused by the surveys, the population surveyed, the timing of the surveys and the results, this part of the study is the least significant. I would probably leave most of this out.

The crux of this study is whether the interventions described (provision of an educational slide set and pocket cards to residents as well as increasing feedback sessions from monthly to bi-monthly) lead to better compliance with meeting all the elements of a resuscitation bundle (whatever that bundle is – needs to be defined). With such low numbers of patients it is unlikely that any difference in any of the other outcome measures listed could possibly reach statistical significance.
I am not confident that there really is a difference in outcome based on the data presented. The following are two of the main reasons:
1. How the outcome is measured is not described. Unless we know how the outcome is measured we cannot be confident that it is measured properly or consistently or objectively
2. There are multiple sources of bias. Some of these are as follows: Firstly, the time periods are separated by 1 year. It is conceivable that other things could have changed in the MICU regarding how patients with severe sepsis/septic shock are treated that may influence the outcome irrespective of the interventions described. Secondly there may be bias in determining the primary outcome as described above. Thirdly, there may be significant differences between the pre and post-intervention resident groups.

---

## Round 0.2 · Minor Revisions

Thank you for submission of the revised manuscript and your constructive response to the reviewer's critique. The revised manuscript has been greatly improved. However, to maintain the rigour of the statistical analysis I would encourage you to amend the last para of Results (line 235): "The 30-day mortality was 25% in the pre-intervention period and improved to 12% in
236 the post-intervention period (p=0.14) (Table 5)." and remove the word "improved" as the data are not statistically significant.

---

## Round 0.3 · accepted · Accept

I am pleased to advise that this contribution has been accepted for publication.